# Design of Robust FEP Porous Ultrafiltration Membranes by Electrospinning-Sintered Technology

**DOI:** 10.3390/polym14183802

**Published:** 2022-09-11

**Authors:** Kaikai Chen, Haoyang Ling, Hailiang Liu, Wei Zhao, Changfa Xiao

**Affiliations:** 1Fiber Materials Research Center, School of Textiles and Fashion, Shanghai University of Engineering Science, Shanghai 201620, China; 2CAS Key Laboratory of Bio-Inspired Materials and Interfacial Science, Technical Institute of Physics and Chemistry, Chinese Academy of Sciences, Beijing 100190, China; 3School of Materials Science and Engineering, State Key Laboratory of Separation Membranes and Membrane Processes, Tiangong University, Tianjin 300387, China

**Keywords:** poly(tetrafluoroethylene-co-hexafluoropropylene) (FEP), ultrafine fibrous membrane, electrospinning-sintered, vacuum membrane distillation

## Abstract

Perfluoropolymer membranes are widely used because of their good environmental adaptability. Herein, the ultrafine fibrous FEP porous membranes were fabricated with electrospinning-sintered technology. The effects of PVA content and sintering temperature on the fabricated membranes’ morphologies and properties were investigated. The results indicate that a kind of dimensionally stable network structure was formed in the obtained ultrafine fibrous FEP porous membranes after sintering the nascent ultrafine fibrous FEP/PVA membranes. The optimal sintering conditions were obtained by comparing the membranes’ performance in terms of membrane morphology, hydrophobicity, mechanical strength, and porosity. When the sintering temperature was 300 °C for 10 min, the porosity, water contact angle, and liquid entry pressure of the membrane were 62.7%, 124.2° ± 2.1°, and 0.18 MPa, respectively. Moreover, the ultrafine fibrous FEP porous membrane at the optimal sintering conditions was tested in vacuum membrane distillation with a permeate flux of 15.1 L·m^−2^·h^−1^ and a salt rejection of 97.99%. Consequently, the ultrafine fibrous FEP porous membrane might be applied in the seawater desalination field.

## 1. Introduction

The electro-spinning of nanofibers has been known since the 1930s [1]. This technique acts as a simple and versatile method that can fabricate fibers in the submicron in nanorange by an electrically charged jet of polymer solution/melt. The extensive polymers and blends also can be used to yield nanofibers [2,3,4]. Moreover, the electro-spinning differs from conventional fiber spinning that can produce cost-effective, highly porous non-woven nanofibrous membrane [5,6]. Commonly used membrane polymers such as cellulose acetate (CA) [7], polysulfone (PSf) [8], and polyvinylidenefluoride (PVDF) [9] have been successfully electro-spun to form non-woven nanofiber membranes for air and water filtrations. For example, investigations have revealed electro-spun nanofibrous membranes possess high flux rates and low transmembrane pressure [10], and hence making them potentially attractive filters in separation technology. These attractive characteristics are attributed to its (1) high porosity, (2) interconnected open pore structure, and (3) tailorable membrane thickness. Moreover, these characteristics are essential for various practical applications such as tissue engineering scaffolds [11,12,13], drug delivery [14], enzyme immobilization [15], battery membrane [16], and filtration materials [17].

Many studies are now utilizing nanoparticles and nanofibers to impart additional properties and functionalities to the membrane and also for membrane preparation and modification. Recently, many review articles in the literature have been dedicated to the application of nanotechnology to water purification [18,19,20]. Subramanian and Seeram [21] reported the developments on the use of electro-spun nanofibers for desalination application by nanofiltration (NF) and membrane distillation (MD). Feng et al. [22] reviewed the preparation and characterization of electro-spun nanofiber membranes for water treatment and other membrane separation processes. Leonard et al. [23] have recently published a comprehensive review of the use of electro-spun nanofibrous membranes for MD application. Furthermore, Zhou et al. [24] have studied vacuum membrane distillation (VMD) using polytetrafluoroethylene (PTFE) nanofiber membranes.

Recently, much research has begun to focus on the perfluoro polymer of poly(tetrafluoroethylene-co-hexafluoropropylene) (FEP), which is a random copolymer of tetrafluoroethylene (TFE) and hexafluoropropylene (HFP), including about 15 wt.% HFP [25]. As PTFE, FEP maintains the exceptional combination of outstanding thermal and chemical resistance and strong hydrophobicity owing to its perfluoro structure [26,27,28,29]. In our previous works [30], we have successfully fabricated the FEP hollow fiber membrane using the melt spinning method. However, no reports of fabricating ultrafine fibrous FEP porous membranes can be found in the past literature.

In this research, the ultrafine fibrous FEP porous membranes were fabricated with the electro-spinning process for the first time. Effects of FEP/polyvinyl alcohol (PVA) mass ratio and sintering temperature on the fabricated membranes’ morphologies and properties were investigated. The obtained ultrafine fibrous FEP porous membranes were applied in a VMD process with a permeate flux of 15.1 L·m^−2^·h^−1^ and a salt rejection of 97.99%, which exhibited a good application prospect in the field of MD or other membrane contactors (MC).

## 2. Experimental

### 2.1. Materials and Chemicals

FEP emulsion (DS603A, solid content is 50 wt.%, average diameter of FEP resin is 250 nm) was supplied by Huaxia Shenzhou New Material Co., Ltd., Zibo, China, and PVA power (1788) was purchased from Hangzhou Lanbo Industrial Co., Ltd, Hangzhou, China. Materials in this experiment were applied without further purification.

### 2.2. Fabrication of Ultrafine Fibrous FEP Porous Membranes

#### 2.2.1. Preparation of Electro-Spinning Solution

The PVA aqueous solution was prepared by dissolving PVA powder in deionized water at 75 °C under constant stirring for at least 6 h. When the solution was cooled down to room temperature, a series contents of FEP emulsion was added to PVA solution with constant stirring for 4 h to form electro-spinning solutions. FEP/PVA mass ratios were 10:1, 8:1, 6:1, and 4:1, respectively, with the same solid concentration of 26 wt.%.

#### 2.2.2. Preparation of Nascent Ultrafine Fibrous FEP/PVA Membranes

Nascent ultrafine fibrous FEP/PVA membranes were fabricated using a typical electro-spinning setup (Yizheng Wanjia Industrial Co., Ltd., Changsha, China) (Figure 1). Basically, the prepared FEP/PVA electro-spinning solution in a tube was pushed slowly into high-voltage-charged sprayers by a syringe pump with a speed 0.06 mL/h. A direct current voltage of 20 kV was applied across a distance of 15 cm between the tip of the sprayers and the grounded rotating collector which was covered by aluminum foil with a speed of 1000 rpm. During the process, ultrafine fibers were produced and collected on the rotating collector. The nascent ultrafine fibrous FEP/PVA membranes were carefully separated from the aluminum foil after steadily spinning for 4 h. Subsequently, membranes were placed in a drying oven under vacuum condition at 60 °C for over 12 h to ensure desiccation.

#### 2.2.3. Sintering Process of Nascent Ultrafine Fibrous FEP/PVA Membranes

The obtained nascent ultrafine fibrous FEP/PVA membranes were fixed in stainless steel plate, and then sintered in a muffle furnace. The furnace was heated to the target temperature with a heating rate of 10 °C/min. Since the melting point of FEP is about 256 °C., temperatures 260 °C, 280 °C, 300 °C, and 320 °C were chosen in this study, and the sintering time was 10 min at each temperature. During the sintering process, nitrogen atmosphere was maintained until the temperature was back to room temperature. Finally, the ultrafine fibrous FEP porous membranes were obtained.

### 2.3. Membrane Properties and Characterization

#### 2.3.1. Morphology of Ultrafine Fibrous FEP Porous Membranes

Scanning electron microscopy (SEM, Hitachi S-4800, Tokyo, Japan)was applied to investigate the morphologies of ultrafine fibrous FEP porous membrane samples. Ultrafine fibrous FEP porous membranes were immersed in liquid N_2_ for 30 s and fractured. Then, samples were all coated in gold and tested in SEM. The distribution of fiber diameters was calculated in SEM images by Image Proplus software.

#### 2.3.2. Differential Scanning Calorimeter (DSC)

The thermal properties of ultrafine fibrous FEP porous membranes were carried out using a Perkin Elmer DSC-7. The calorimeter operated under nitrogen atmosphere. Membrane samples weighing about 6 mg closed in aluminum pans were heated from up to 350 °C at 10 °C/min and then cooled to room temperature at the same rate. The crystallinity value (*X_c_*) was calculated from the following Equation (1):(1)$$X_{c}=\frac{\Delta H_{m}}{\Delta H_{m100}}\times 100\%$$
where Δ*H_m_* and Δ*H*_*m*100_ (87.9 J/g) represent the melting enthalpy of the investigated samples and 100% crystalline FEP, respectively.

#### 2.3.3. Water Contact Angle (WCA)

The WCA of all the samples was measured by an optical contact angle meter (DCAT11, Dataphysics, Filderstadt, Germany), model JYSP-180) at room temperature. The diameter of the water droplet was about 1 mm, lasting for 10 s after the droplet was dropped on the sample surface by vibrating the tip of a micro-syringe. A lens and a source light were used to create the drop image on a screen. The WCA was determined with the projected drop image. Five different spots for each sample were measured. The average value of the five spots as the WCA was chosen.

#### 2.3.4. Liquid Entry Pressure (LEP)

The optimal sintering condition was chosen to fabricate ultrafine fibrous FEP porous membranes according to the membrane properties. The LEP of dried ultrafine fibrous FEP porous membranes was accessed using a laboratory device (Figure 2) at room temperature. The pressure was slowly increased until the water seeped out, and the value of pressure gauges was the LEP. The average value of three tests as the LEP was chosen.

#### 2.3.5. Nitrogen Flux through the Membranes

Nitrogen flux of dry ultrafine fibrous FEP porous membranes was measured by a laboratory device (Figure 3) by the following Equation (2), and the permeate flow rate was measured at a pressure of 0.1 MPa.
(2)$$J=\frac{L}{A}$$
where *J* is the nitrogen flux (m^3^·m^−2^·h^−1^), *L* is the nitrogen flow (m^3^·h^−1^), and *A* is the membrane area (m^2^).

#### 2.3.6. Porosity and Pore Size Distribution

The gravimetric method was used for assessing the porosity by calculating the weight of liquid immersed in the membrane pores. Owing to the strong hydrophobicity of FEP, n-butyl alcohol was used as the wetting liquid. The ultrafine fibrous FEP porous membrane samples were immersed in the n-butyl alcohol for at least 24 h. The n-butyl alcohol of the membrane surface was removed by a filter paper. After that, the wet membrane’s weight was measured. Additionally, the dry membranes’ weight was measured after drying in an electric blast drying oven for 10 h at the temperature of 30 °C. The porosity (*ε*) was calculated by Equation (3) [22,31]:(3)$$\varepsilon (\%)=\frac{W_{1}-W_{2}}{Ad\rho }\times 100\%$$
where *A* is the area of the membrane (mm^2^), *d* is the average thickness of the membrane (mm), *ρ* is the n-butyl alcohol density (*ρ* = 0.811 g/mL), *W*_1_ is the weight of wet membrane (g), and *W*_2_ is the weight of the dry membrane (g).

The pore size distribution of ultrafine fibrous FEP porous membranes was investigated by using a Capillary Flow Porometer (CFP-1100-A*, Newtown Square, PA, United States). The membranes were fully wetted with the wetting liquid, and then the measurements were carried out following the procedure described in the literature [32]. The pore size distribution was determined with the aid of the computer software coupled to the capillary flow porometer.

#### 2.3.7. Mechanical Strength

The mechanical properties of ultrafine fibrous FEP porous membranes were measured by YG-061F electronic single yarn tensile tester (Yantai, China), and 2 mm/min was used for the tensile rate. The average measurement of the five specimens was used.

### 2.4. VMD Experiment

Experiments on Vacuum Membrane Distillation (VMD) were carried out to evaluate the permeate performance of ultrafine fibrous FEP porous membranes. The desalination experiment was performed using a setup schematically shown in Figure 4. One side of the membrane was in contact with a hot, circulating salt solution, and its other side was connected to a vacuum pump to withdraw the permeated water vapor. The water vapor was subsequently condensed into liquid water by a glass condenser using tap water as coolant. The condensed water was collected in a glass bottle, and its volume was determined with a measuring cylinder. The conductivity of the feed solution and permeate water was measured by a conductivity meter (AP-2, HM). The NaCl rejection R was calculated by the following Equation (4):(4)$$R=\left(1-\frac{C_{p}}{C_{f}}\right)\times 100\%$$
where *C_f_* and *C_p_* were the conductivities of the feed solution and permeate water, respectively.

## 3. Results and Discussion

### 3.1. Membrane Morphology

#### 3.1.1. Effects of FEP/PVA Mass Ratios

As mentioned above, the four different mass ratios of FEP/PVA (10:1, 8:1, 6:1, 4:1) were investigated in this paper. The surface morphologies of the obtained nascent ultrafine fibrous FEP porous membranes are shown in Figure 5. Owing to its insolubility in common solvents, the pure FEP could not be electro-spun into ultrafine fibers. In order to obtain nascent ultrafine fibrous FEP porous membranes, a subtractive matrix polymer and post-treatment were introduced into the process of fabricating nascent ultrafine fibrous FEP porous membranes. PVA, a water-soluble polymer, exhibits good spinnability, and it can be electro-spun into ultrafine or nano fibers easily. It was demonstrated that the nanofibers of the chitosan, hydroxyapatite, and zinc oxide were electro-spun with PVA as membrane carrier [33,34,35]. It can be found in Figure 5(A1,B1,C1,D1) that with the increasing content of PVA, nascent ultrafine fibrous FEP porous membranes obviously transformed from the beadlike structure to the fibrous structure. When the mass ratio of FEP/PVA was 10:1 (Figure 5(A1)), only a beadlike structure was obtained, while the FEP/PVA mass ratio reached 6:1 (Figure 5(C1)), and fibers of about 500 nm in diameter were formed. While increasing the PVA mass ratio further, the fiber diameters were increased. The statistics of fiber diameters were illustrated in Figure 5. The fiber diameters fluctuated in the range between 300 and 700 nm. In order to obtain nascent ultrafine fibrous porous membranes with high FEP content, PVA content should be reduced as possible on the premise of good spinnability. In this study, a FEP/PVA mass ratio of 6:1 was chosen for further investigation due to the uniform fiber diameter and pore structure.

#### 3.1.2. Effects of Sintering Temperature

The SEM images of ultrafine fibrous FEP porous membranes sintered at different temperatures were shown in Figure 6. From the SEM images, it can be clearly observed that the FEP particles gradually fused during the sintering process. The nascent ultrafine fibrous FEP/PVA membranes were assembled by random ultrafine fibers. The ultrafine fibers showed an interconnected fibrous network in the images (Figure 6B–D). As the sintering temperature increased, the FEP resins fused with each other furtherly, which induced not only lower membrane porosity but also smaller pore size. However, the mechanical strength of the membrane improved. Therefore, the sintering temperature is a very important factor that endows the membrane with suitable porosity and favorable mechanical strength. The membrane samples were obtained at the sintering temperature of 300 °C for 10 min. A dimensionally stable network structure was formed. Figure 6E showed the membrane which was treated at 320 °C. The fibers fused together, and the membrane presented a compact structure. Digital photos of ultrafine fibrous FEP porous membranes sintered at different temperatures are shown in Figure 6(A2,B2,C2,D2). The color of the membranes became deeper with the increase in sintering temperature.

### 3.2. DSC Analysis

Figure 7 shows the typical differential scanning calorimeter (DSC) curves of ultrafine fibrous FEP porous membrane samples, and the corresponding data were tabulated in Table 1. As shown in the heating curves (Figure 7A), there was a endothermic peak at 87.5 °C of nascent ultrafine fibrous FEP/PVA membranes. Moreover, the endothermic peak disappeared after sintering. These results indicate that the PVA was totally decomposed during the sintering process. The melting temperature of ultrafine fibrous FEP porous membranes increased with the increase in sintering temperature. Meanwhile, the enthalpy and the degree of crystallinity (X_c_) increased. As for the cooling curves (Figure 7B), the crystallization peak moved towards the lower temperatures with the increase in sintering temperature. These results should be attributed to the nascent electro-spun fibers being randomly distributed and not interconnected. The sintering process enhanced the dimensional integrity and mechanical properties of the membranes.

### 3.3. WCA Analysis

During the MD process, hydrophobicity is one of the most important factors. As we know, the pore size, surface roughness, and composition of the membrane were the main factors to decide WCA [36]. The WCA of sintered membranes at different temperatures are shown in Figure 6 and Table 2. Due to the large pore size and PVA in the nascent ultrafine fibrous FEP/PVA membranes, it is easy to absorb a water drop into the membrane completely. However, the ultrafine fibrous FEP porous membranes exhibited strong hydrophobicity, and the WCA value increased with the increase in sintering temperature, which resulted in higher LEP and salt rejection of the MD application.

### 3.4. Permeability

The porosity, N_2_ flux, and LEP of the ultrafine fibrous FEP porous membranes are tabulated in Table 3. As analyzed above, increased sintering temperature would induce not only lower membrane porosity but also smaller pore size, which displayed a decrease in porosity and N_2_ flux. It was due to the fact that the network structure became denser with the increase in sintering temperature as shown in Figure 6. Meanwhile, membrane thickness was also another factor that influenced the permeability of the fibrous membrane. Obvious differences in the thicknesses were observed among the ultrafine fibrous FEP porous membranes with different sintering temperature as listed in Table 3. The porosity of ultrafine fibrous FEP porous membranes reduced to about 60% when the sintering temperature increased above 300 °C.

N_2_ flux and LEP are two important membrane characteristics for MD which could provide a high MD flux. From Table 3, it can be seen that the increase in sintering temperature improved the LEP value, while the N_2_ flux decreased. These results should be attributed to the reduction in pore sizes and the improved hydrophobicity because of the structure of the ultrafine fiber assembling.

In this study, the ultrafine fibrous FEP porous membranes were also preferred for the MD process at the sintering temperature of 300 °C for 10 min.

### 3.5. Pore Size Distribution

The pore size distribution is a crucial parameter of performance during the MD process [27]. As stated by Schofield et al., the membranes utilized in MD should have a reasonably small pore size (be preferably smaller than 0.5 μm) to prevent wetting [37]. Figure 8 showed the pore size distribution curves of the ultrafine fibrous FEP porous membranes prepared from different sintering temperature. It can be found that the pore sizes of the ultrafine fibrous FEP porous membranes become smaller with the increase in sintering temperature, which agreed well with the results of SEM. The membrane sintered at 260 °C showed a broad pore size distribution ranging from 0.5 to 5.1 µm. This was due to the large fibrous network of the electro-spun fibers, as discussed in the SEM results above. When the sintering temperature was 280 °C, the pore size distribution of the ultrafine fibrous FEP porous membranes became narrower (Figure 8B). Therefore, it could be concluded that a higher sintering temperature tended to result in a smaller pore size and narrower pore size distribution (Figure 8C,D).

### 3.6. Mechanical Strength

Figure 9 shows the stress–strain curves of the ultrafine fibrous FEP porous membranes. It can be seen that the mechanical properties of the membrane samples improved significantly after sintering. These results can be explained by the structural and compositional changes at different sintering temperature. The nascent membranes were formed by multiple layers of randomly oriented composite ultrafine fibers of FEP particles in PVA matrix. However, the microstructure changed, and the FEP particles were almost completely fused when the sintering temperature was up to 320 °C. These results suggest that the changes in the microstructure observed at 260 °C was due to the melting of FEP particles, and furthermore, it also suggests that the increase in tensile strength of the ultrafine fibrous FEP porous membranes was related to the melting of FEP particles. All the changes are shown in the SEM images (Figure 6).

### 3.7. VMD Experiment

According to the results above, we chose the optimal sintering conditions by overall consideration of the WCA, porosity, LEP value, and mechanical strength. In the following experiments, ultrafine fibrous FEP porous membranes were prepared by sintering the nascent ultrafine fibrous FEP/PVA membranes at 300 °C for 10 min.

In the VMD process, one of the main factors is the vacuum pressure [35]. Thus, the conductivity and permeate flux were carried out by the effect of vacuum pressure. It could be seen from Figure 10 that increase in vacuum pressure induced the improvement of permeate flux. The permeate flux reached as high as 15.1 L·m^−2^·h^−1^ when vacuum pressure was 0.06 MPa while feed temperature was 80 °C, and the salt rejections achieved 97.99%, which indicated that the obtained ultrafine fibrous FEP porous membranes exhibited good application prospects in the field of MD.

Meanwhile, the effects of feed temperature are exhibited in Figure 11 with the feed temperature heated from 65 °C to 80 °C at the inlet of the membrane module. As the feed temperature increased, there was enhancement of the permeate flux in all ultrafine fibrous FEP porous membranes. It was attributed to the effect of increased feed temperature, which resulted in more mass transfer. The salt rejections of the samples also achieved 97.93%.

### 3.8. Comparison with Other VMD Membranes

The results of comparisons with other MD processes are listed in Table 4. It can be observed that the permeability in this study was comparable or even better than most of the previous reports. This may be due to the fact that the porosity of the membranes was higher, and pore size was suitable for MD. Compared with ultrafine fibrous PTFE porous membrane [38], the comprehensive performances of fibrous FEP porous membrane were slightly low. However, the lower sintering temperature and lower preparation cost were the advantages of the fibrous FEP porous membrane. Moreover, owing to the excellent chemical and thermal resistance of FEP than PVDF [38,39,40], it is believed that there would be good application prospects in the field of MD, especially in systems with high temperature, acid, alkali, and other harsh environments.

## 4. Conclusions

The ultrafine fibrous FEP porous membranes used for VMD were fabricated by the electrospinning-sintering method. The optimal conditions were chosen so that the FEP/PVA mass ratio was 6:1, and sintering temperature was 300 °C for 10 min, while comparing with other preparation conditions. The membrane’s thickness, porosity, WCA, and LEP were 82 μm, 62.7%, 124.2° ± 2.1°, and 0.18 MPa, respectively, which was applied in the VMD process. The permeate flux reached as high as 15.1 L·m^−2^·h^−1^ when trans-membranous pressure was 0.06 MPa and feed temperature was 80 °C, and the salt rejections also achieved 97.99% when the feed NaCl concentration was 3.5 wt.%. Our preliminary assessment of ultrafine fibrous FEP porous membranes showed that this method has a high potential to fabricate MD membranes for desalination processes. This is of great significance for energy saving and purification in the field of seawater desalination.

## Figures and Tables

**Figure 1 polymers-14-03802-f001:**
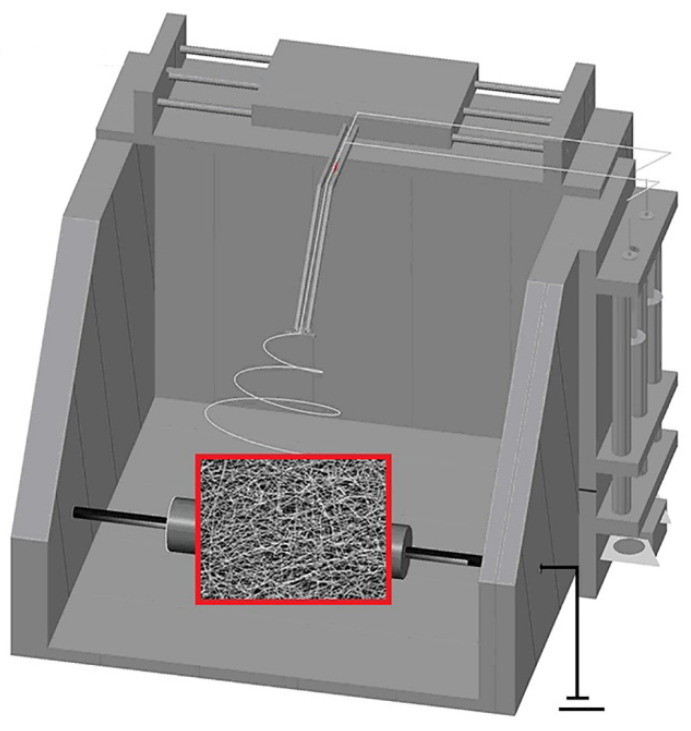
The schematic diagram of electro-spinning apparatus.

**Figure 2 polymers-14-03802-f002:**
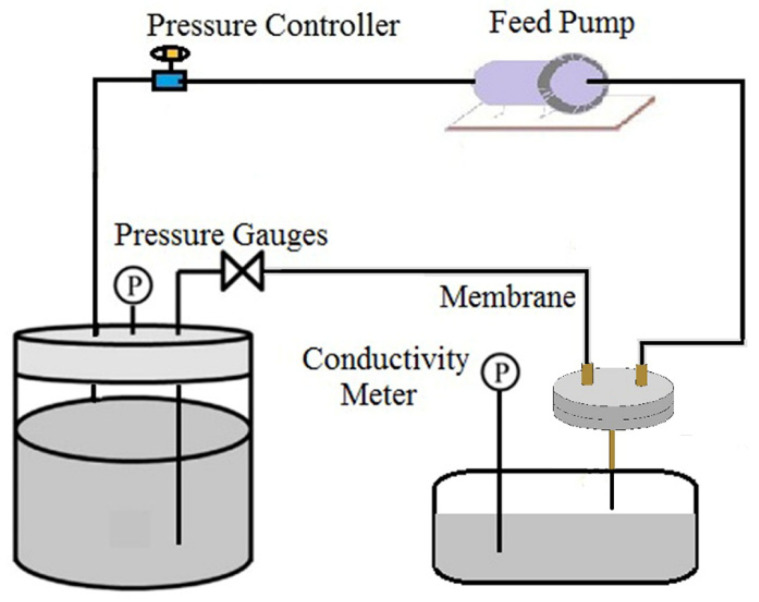
The testing device for the LEP of the dried ultrafine fibrous FEP porous membranes.

**Figure 3 polymers-14-03802-f003:**
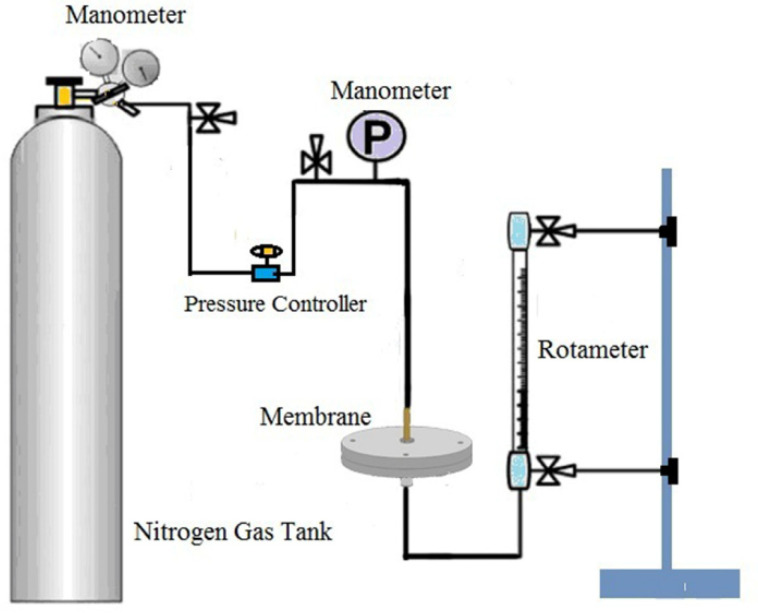
The testing device for the nitrogen flux of dried ultrafine fibrous FEP porous membranes.

**Figure 4 polymers-14-03802-f004:**
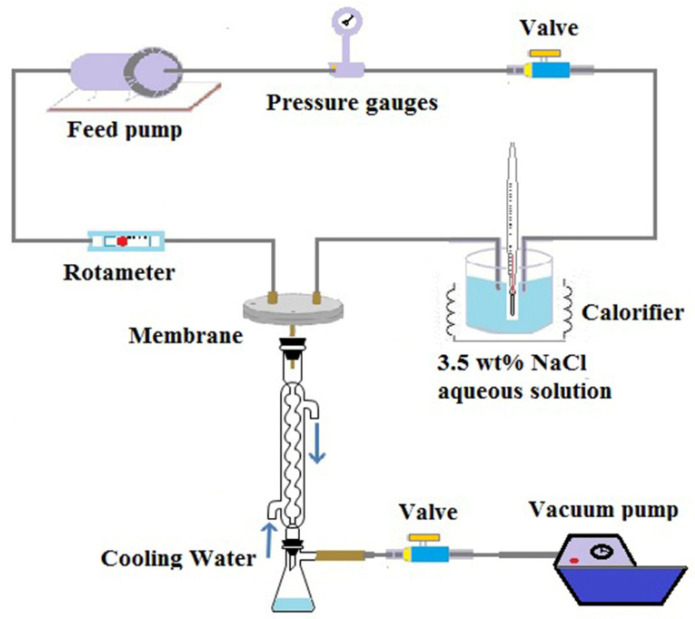
Schematic diagram of the experimental VMD apparatus.

**Figure 5 polymers-14-03802-f005:**
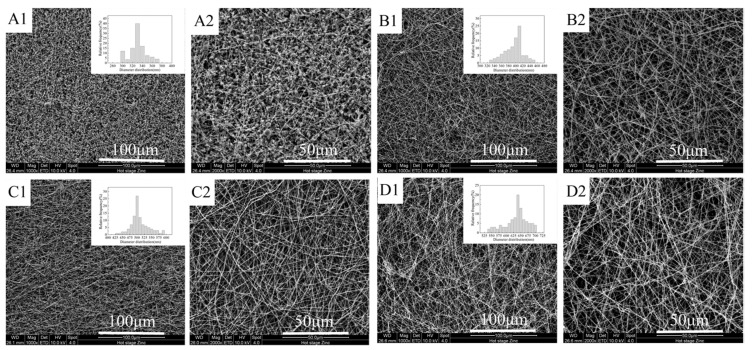
Surface morphologies of nascent ultrafine fibrous FEP porous membranes at different FEP/PVA mass ratios (**A**): 10:1; (**B**): 8:1; (**C**): 6:1; (**D**): 4:1; 1: 1000× surface; 2: 2000× surface).

**Figure 6 polymers-14-03802-f006:**
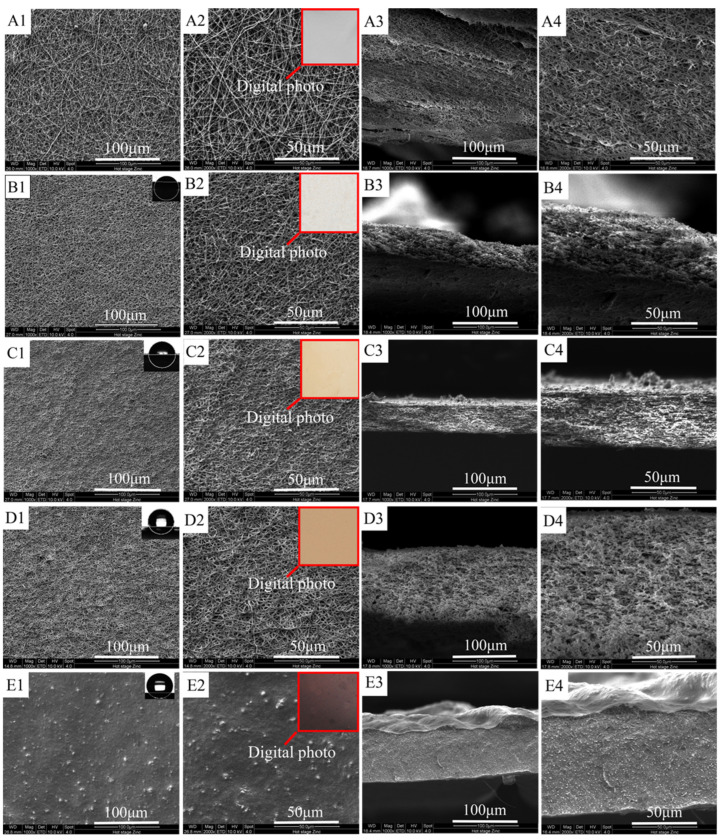
Morphologies of ultrafine fibrous FEP porous membranes at different sintering temperature (FEP/PVA mass ratio 1:6; (**A**): nascent membrane; (**B**): 260 °C; (**C**): 280 °C; (**D**): 300 °C; (**E**): 320 °C; 1: 1000× surface; 2: 2000× surface; 3: 1000× cross-section; 4: 2000× cross-section).

**Figure 7 polymers-14-03802-f007:**
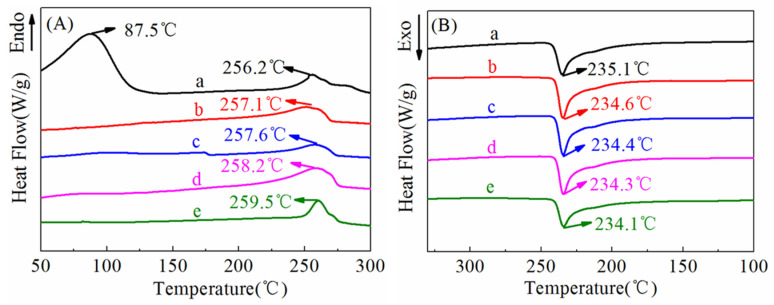
DSC curves of ultrafine fibrous FEP porous membranes at different sintering temperature (FEP/PVA mass ratio: 1:6; a: nascent membrane; b: 260 °C; c: 280 °C; d: 300 °C; e: 320 °C; (**A**)-heating curve, (**B**)-cooling curve).

**Figure 8 polymers-14-03802-f008:**
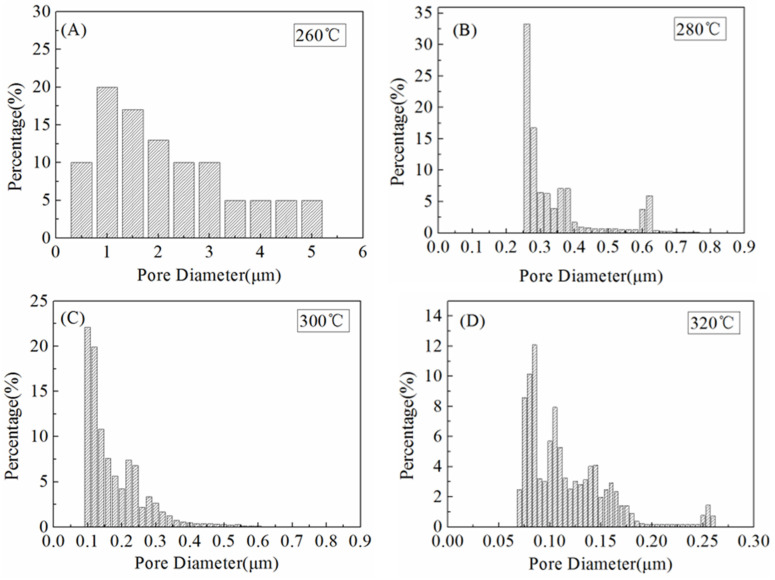
Pore size distribution of the ultrafine fibrous FEP porous membranes at different sintering temperature (FEP/PVA mass ratio: 1:6). (**A**) 260 °C; (**B**) 280 °C; (**C**) 300 °C; (**D**) 320 °C

**Figure 9 polymers-14-03802-f009:**
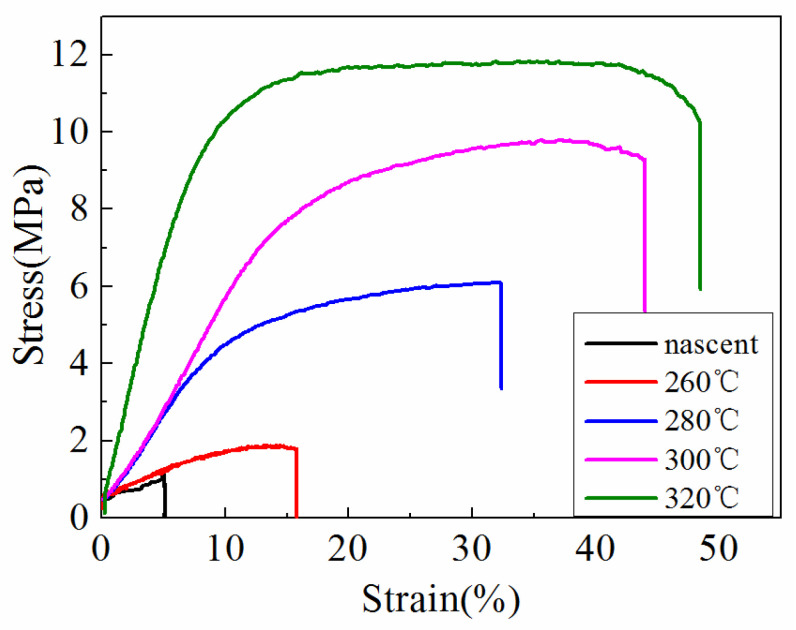
Stress–strain curves of the ultrafine fibrous FEP porous membranes at different sintering temperature (FEP/PVA mass ratio: 1:6).

**Figure 10 polymers-14-03802-f010:**
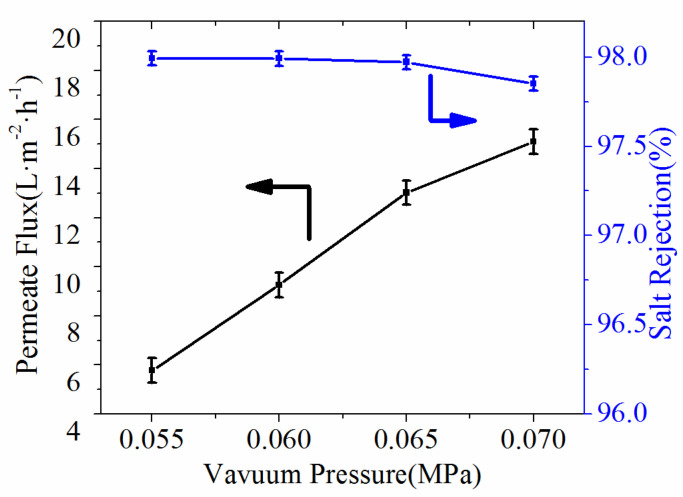
Effect of vacuum pressure on the permeate flux and salt rejection (NaCl concentration, 3.5 wt.%; feed temperature, 80 °C).

**Figure 11 polymers-14-03802-f011:**
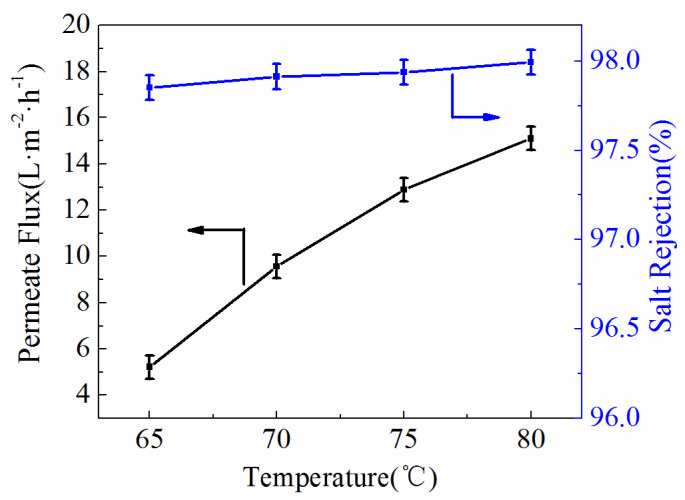
Effect of feed temperature on the permeate flux and salt rejection (NaCl concentration, 3.5 wt.%; Vacuum pressure, 0.07 MPa).

**Table 1 polymers-14-03802-t001:** Thermal property of ultrafine fibrous FEP porous membranes by DSC.

Sintering Temperature (°C)	*T_m_* (°C)	*T_c_* (°C)	*ΔH_m_* (J/g)	*X_c_* (%)
-	256.2	235.1	27.4	31.2
260	257.1	234.6	35.9	40.9
280	257.6	234.4	39.1	44.5
300	258.2	234.3	40.9	46.6
320	259.5	234.1	45.2	51.8

**Table 2 polymers-14-03802-t002:** The data of water contact angle (n = 5).

Sintering Temperature (°C)	-	260	280	300	320
WCA (°)	0	58.9 ± 1.9	88.8 ± 1.8	124.2 ± 2.1	131.9 ± 1.7

**Table 3 polymers-14-03802-t003:** The characteristics of the ultrafine fibrous FEP porous membranes.

	Sintering Temperature (°C)	280	300	320
Thickness (μm)	43 ± 5	48 ± 2	82 ± 3	86 ± 2
Porosity (%)	96.1 ± 1	84.0 ± 2	62.7 ± 1	31.5 ± 1
N2 flux (m^3^·m^−2^·h^−1^)	-	-	20.2 ± 3	1.37 ± 2
LEP (MPa)	-	0.04	0.18	0.32

**Table 4 polymers-14-03802-t004:** Permeate flux in this study compared with other membranes in VMD processes.

Membrane Code	Porosity (%)	Feed Solution	Feed Temperature (°C)	Vacuum (MPa)	Permeate Flux (L·m^−2^·h^−1^)	Reference
PVDF flat-sheet	-	3.5 wt.% NaCl	30	0.05	12.6	[38]
PTFE flat-sheet	60.0	5 wt.% Acetone	80	0.07	12.5	[30]
PTFE ultrafine fiber	79.8	3.5 wt.% NaCl	80	0.03	15.9	[40]
FEP ultrafine fiber	62.7	3.5 wt.% NaCl	80	0.06	15.1	This study

## Data Availability

The data presented in this study are available on request from the corresponding author.

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
