# Peer review of "Design of Robust FEP Porous Ultrafiltration Membranes by Electrospinning-Sintered Technology"

_polymers, 2022, doi:10.3390/polym14183802_

Round 1

Reviewer 1 Report

This paper studies the electro-spinning sintering of one perfluoropolymer membrane, FEP. It studies the effects of PVA (polyvinyl alcohol) content and sintering temperature on the morphology and different properties (hydrophobicity, mechanical strength, and porosity) of the membranes. The application was in vacuum membrane distillation (VMD).

This is a quite typical paper on lab scale experimenting on a new process, or actually a modification of the existing process. It has a proper structure for such a paper and detailed handling of the results. Some comments and proposals for improving the paper:

Experiment design: (1) It is told that four different FEP/PVA mass ratios and four temperatures were tested. It seems that the authors have started by testing FEP/PVA ratio at four levels, used the “best” ratio and continued testing with temperature at four levels. But, what if these variables had some interaction, and some other combination had given even better results?

Optimal sintering conditions: (2) The authors speak about “optimized membranes” on lines 216-8 and return to this topic on lines 305-8 where they list the parameters considered in finding “the optimal sintering conditions”. I seems that they decide that 300 ÌŠC for 10 minutes is the optimum, probably with the FEP/PVA ratio of 1:6. They have made all tests with 10 minutes sintering time, so there is no valid evidence on the optimality of this time. The topic of principles for defining optimal conditions and the parameters effecting on it deserve a more detailed discussion in Section 2.

(3) Conclusion needs few sentences about the further research directions.

(4) Language is reasonable, but at least following mistakes should be corrected:

line 96: and then sintered in the muffle furnace

97-8: Since the melting point of FEP is about 256℃, temperatures 260℃, 280℃, 300℃, 320℃ were chosen in this study

117: represent the melting enthalpy

131: This sentence is unclear: Increase the pressure slowly until the mutation of the conductivity meter.

134: The testing device for LEP of the dried ultrafine fibrous FEP porous membranes

139: where, J is the nitrogen flux (m3·m-2·h-1), L is the nitrogen flow (m3·h-1), and A is … See also  lines 152-4.

142: The testing device for …

144: by calculating

163-4: The average measurement of the five specimens was used..

165: Vacuum Membrane Distillation

168: One side of the

196: While increasing the PVA mass ratio further,

201-2: for further investigation.

226: DSC analysis

243: WCA analysis

Author Response

Reviewer: #1

This paper studies the electro-spinning sintering of one perfluoropolymer membrane, FEP. It studies the effects of PVA (polyvinyl alcohol) content and sintering temperature on the morphology and different properties (hydrophobicity, mechanical strength, and porosity) of the membranes. The application was in vacuum membrane distillation (VMD).

This is a quite typical paper on lab scale experimenting on a new process, or actually a modification of the existing process. It has a proper structure for such a paper and detailed handling of the results. Some comments and proposals for improving the paper:

Technical Aspects:

  1. Comment: Experiment design: (1) It is told that four different FEP/PVA mass ratios and four temperatures were tested. It seems that the authors have started by testing FEP/PVA ratio at four levels, used the “best” ratio and continued testing with temperature at four levels. But, what if these variables had some interaction, and some other combination had given even better results?

Response: Considering the Reviewer’s suggestion, we have illustrated the interaction and combination of these variables in Section 2.2.1. The revised details were as following:

The PVA aqueous solution was prepared by dissolving PVA powder in deionized water at 75℃ under constant stirring for at least 6 h. When the solution was cooled down to room temperature, a series contents of FEP emulsion was added to PVA solution with constant stirring for 4 h to form electro-spinning solutions. The FEP/PVA mass ratios were 10:1, 8:1, 6:1, and 4:1 respectively with the same solid concentration of 26 wt.%.

  1. Comment: Optimal sintering conditions: (2) The authors speak about “optimized membranes” on lines 216-8 and return to this topic on lines 305-8 where they list the parameters considered in finding “the optimal sintering conditions”. I seems that they decide that 300 ÌŠC for 10 minutes is the optimum, probably with the FEP/PVA ratio of 1:6. They have made all tests with 10 minutes sintering time, so there is no valid evidence on the optimality of this time. The topic of principles for defining optimal conditions and the parameters effecting on it deserve a more detailed discussion in Section 2.

Response: As the reviewer suggested, we have revised the principles for defining optimal conditions and the parameters so as to make it clear. The revised details were as following:

  1. The obtained nascent ultrafine fibrous FEP/PVA membranes were fixed in stain-less steel plate, and then sintering in muffle furnace. The furnace was heated to the target temperature with a heating rate of 10℃/min. Since the melting point of FEP is about 256℃. In this study, 260℃, 280℃, 300℃, 320℃ were chosen and the sintering time was 10min at each temperature. During the sintering process, nitrogen atmos-phere was maintained until the temperature was back to room temperature. Finally, the ultrafine fibrous FEP porous membranes were obtained.
  2. As mentioned above, the four different mass ratios of FEP/PVA (10:1, 8:1, 6:1, 4:1) were investigated in this paper. The surface morphologies of the obtained nascent ul-trafine fibrous FEP porous membranes were shown in Fig. 5. Owing to its insolubility in common solvents, the pure FEP could not be electro-spun into ultrafine fibers. In order to obtain nascent ultrafine fibrous FEP porous membranes, a subtractive matrix polymer and post-treatment were introduced into the process of fabricating nascent ul-trafine fibrous FEP porous membranes. PVA, a water-soluble polymer, exhibits good spinnability and it can be electro-spun into ultrafine or nano fibers easily. It was demonstrated that the nanofibers of the chitosan, hydroxyapatite, and zinc oxide were electro-spun with PVA as membrane carrier[28–30]. It can be found in Fig. 5 [(A1), (B1), (C1) and (D1)], with the increasing content of PVA, nascent ultrafine fibrous FEP po-rous membranes transformed from the beadlike structure to fibrous structure obvious-ly. When the mass ratio of FEP/PVA was 10:1 [Fig. 5(A1)], only beadlike structure was obtained while the FEP/PVA mass ratio reached 6:1 [Fig. 5(A3)], fibers of about 500nm in diameter were formed. As the increased PVA mass ratio further, the fiber diameters were increased. The statistics of fiber diameters were illustrated in Fig. 5. The fiber di-ameters fluctuated in a small and reasonable range between 300 and 700nm. In order to obtain nascent ultrafine fibrous porous membranes with high FEP content, PVA con-tent should be reduced as possible on the premise of good spinnability. In this study, a FEP/PVA mass ratio of 6:1 was chosen for further investigated due to the uniform fiber diameter and pore structure.

  1. Comment: Conclusion needs few sentences about the further research directions.

Response: It is really true as Reviewer suggested that the conclusion needs few sentences about the further research directions. The revised details were as following:

The ultrafine fibrous FEP porous membranes used for VMD were fabricated by electrospinning-sintering method. The optimal conditions were chosen that the FEP/PVA mass ratio was 6:1 and sintering temperature was 300℃ for 10 minutes. The membrane with thickness, porosity, WCA and LEP was 82 μm, 62.7%, 124.2°±2.1°, and 0.18MPa respectively, which was applied in VMD process. The permeate flux reached as high as 15.1 L·m-2·h-1 when trans-membranous pressure was 0.06MPa and feed tem-perature was 80℃, and the salt rejections also achieved 97.99% when the feed NaCl concentration was 3.5 wt.%. Our preliminary assessment of ultrafine fibrous FEP po-rous membranes showed that this method has a high potential to fabricate MD mem-branes for desalination processes. It is of great significance for energy saving and puri-fication in the field of seawater desalination.

  1. Comment: Language is reasonable, but at least following mistakes should be corrected:

Line 96: and then sintered in the muffle furnace

97-8: Since the melting point of FEP is about 256℃, temperatures 260℃, 280℃, 300℃, 320℃ were chosen in this study

117: represent the melting enthalpy

131: This sentence is unclear: Increase the pressure slowly until the mutation of the conductivity meter.

134: The testing device for LEP of the dried ultrafine fibrous FEP porous membranes

139: where, J is the nitrogen flux (m3·m-2·h-1), L is the nitrogen flow (m3·h-1), and A is … See also  lines 152-4.

142: The testing device for …

144: by calculating

163-4: The average measurement of the five specimens was used.

165: Vacuum Membrane Distillation

168: One side of the

196: While increasing the PVA mass ratio further,

201-2: for further investigation.

226: DSC analysis

243: WCA analysis

Response: According to the reviewer’s comments, we have made the linguistic modification, and the revised details were revised in red word in the manuscript. For example, the Line97-8 was revised to “Since the melting point of FEP is about 256℃, temperatures 260℃, 280℃, 300℃”.

Finally, special thanks to you for your good comments.

We have tried our best to improve the manuscript and made some changes in the manuscript. We appreciate for your warm work earnestly and hope that the correction will meet with approval. Thanks a lot for your comments and suggestions.

Reviewer 2 Report

Please find my comments below:

  1. Introductions

Line 29 – 30 – Submicron in nano-range. Please rephrase.

Line 32 – 34 – State the differences between the two. Provide references.

Line 34 – 36 – Provide references.

Line 43 – “…and filtration membrane” already mentioned. Please correct.

Line 44 – 46 – While nanoparticles and nanofibers provide additional properties to the membrane, it is implied that they are used in preparation, fabrication (not the same as preparation?) and modification. Please rephrase.

Line 54 – PTFE provide full name.

Line 55 – 58 – Provide more references for these studies.

Line 61 – Please explain “ultrafine” precisely. What are the dimensions or ranges for ultrafine? Also, has anyone else fabricated “fine” fibrous FEP for membranes or other uses? If so, please provide mentions and references.

Line 66 – “fully” is not necessary.

  1. Experimental

Line 82 – Why did the authors choose 26% for their total solid concentration? Please explain.

Line 89 – I would recommend rpm instead of r/min.

Line 96 – “…and then sintering in…”. Please correct.

Line 131 – 132 – Please rephrase.

Line 136 – I suppose it’s nitrogen flux through the membranes. Please rephrase.

3. Results

Line 196 – There is no Fig. A3.

Line 198 – “small and reasonable” is not really scientific writing. Please rephrase/correct.

Line 208 – 210 – Authors’ claim concerning PVA decomposition is based on SEM images. Please provide a more thorough explanation or provide a more objective proof or remove claim.

Line 216 – Previously it was stated that sintering induced lower porosity. Here authors claim otherwise. Please correct.

Line 216 – Explain “optimized” or remove.

Line 231 – 232 – Please explain the exact relation between the presence or absence of the endothermic 87.5oC peak and the claimed PVA decomposition. In general, I would advise care when making definitive statements and proof or scientific explanation.

Line 249 – Improper use of “deserved”.

Line 335 – Please provide reference or provide more details for the better chemical and thermal resistance of FEP, when compared to PVDF.

Line 335 – “much more excellent” should be replaced with “better”.

4. Conclusion

Line 341 – When discussing “optimal conditions”, please refer to the context.

Author Response

Reviewer: #2

Please find my comments below:

Technical Aspects:

  1. Comment:

Line 29 – 30 – Submicron in nano-range. Please rephrase.

Line 32 – 34 – State the differences between the two. Provide references.

Line 34 – 36 – Provide references.

Line 43 – “…and filtration membrane” already mentioned. Please correct.

Line 44 – 46 – While nanoparticles and nanofibers provide additional properties to the membrane, it is implied that they are used in preparation, fabrication (not the same as preparation?) and modification. Please rephrase.

Line 54 – PTFE provide full name.

Line 55 – 58 – Provide more references for these studies.

Line 61 – Please explain “ultrafine” precisely. What are the dimensions or ranges for ultrafine? Also, has anyone else fabricated “fine” fibrous FEP for membranes or other uses? If so, please provide mentions and references.

Line 66 – “fully” is not necessary..

Response: According to the reviewer’s comments, we have made the linguistic modification, and the revised details were revised in red word in the manuscript. For example, the Line61 was revised as following:

Yes, the ultrafine has been utilized as following references:

[1] Huang Y ,  Huang Q L ,  Liu H , et al. Preparation, characterization, and applications of electrospun ultrafine fibrous PTFE porous membranes[J]. Journal of Membrane Science, 2017, 523:317-326.

[2]Huang, Huang, Qing-Lin. Preparation, characterization, and applications of electrospun ultrafine fibrous PTFE porous membranes[J]. Journal of Membrane Science, 2017(523-).

[3] Shi Z ,  Ju J ,  Liang Y , et al. A Comparative Study of Poly(tetrafluoroethylene) Ultrafine Fibrous Porous Membranes Prepared by Electrospinning, Solution Blowing Spinning, and Electroblown Spinning[J]. Chemistry Letters, 2016:cl.160877.

  1. Comment: Experimental

Line 82 – Why did the authors choose 26% for their total solid concentration? Please explain.

Line 89 – I would recommend rpm instead of r/min.

Line 96 – “…and then sintering in…”. Please correct.

Line 131 – 132 – Please rephrase.

Line 136 – I suppose it’s nitrogen flux through the membranes. Please rephrase.

Response: According to the reviewer’s comments, we have made the linguistic modification, and the revised details were revised in red word in the manuscript. For example, the Line82 was revised as following:

It is really true as Reviewer suggested that 26% for their total solid concentration should be explained. Based on our preliminary experiment in this study, the solid concentration of 26 wt.% was easy to be prepared the nascent ultrafine fibrous FEP/PVA membranes by electrospinning method.

  1. Comment: Results

Line 196 – There is no Fig. A3.

Line 198 – “small and reasonable” is not really scientific writing. Please rephrase/correct.

Line 208 – 210 – Authors’ claim concerning PVA decomposition is based on SEM images. Please provide a more thorough explanation or provide a more objective proof or remove claim.

Line 216 – Previously it was stated that sintering induced lower porosity. Here authors claim otherwise. Please correct.

Line 216 – Explain “optimized” or remove.

Line 231 – 232 – Please explain the exact relation between the presence or absence of the endothermic 87.5℃ peak and the claimed PVA decomposition. In general, I would advise care when making definitive statements and proof or scientific explanation.

Line 249 – Improper use of “deserved”.

Line 335 – Please provide reference or provide more details for the better chemical and thermal resistance of FEP, when compared to PVDF.

Line 335 – “much more excellent” should be replaced with “better”.

Response: According to the reviewer’s comments, we have made the linguistic modification, and the revised details were revised in red word in the manuscript. For example, the Line 208-210 was revised as following:

As reported previous[1-3], the PVA decomposition could be occurred by high temperature sintering. However, it is difficult to find the process in SEM images. Consequently, we delete the expression in the manuscript.

[1] Li C ,  Hou T ,  She X , et al. Decomposition properties of PVA/graphene composites during melting-crystallization[J]. Polymer Degradation & Stability, 2015, 119(sep.):178-189.

[2] Xie W ,  Bao Q ,  Liu Y , et al. Hydrogen Bond Association to Prepare Flame Retardant Polyvinyl Alcohol Film with High Performance[J]. ACS Applied Materials & Interfaces, 2021, 13(4).

[3] [1] Zhou J ,  Lin Y ,  Wang L , et al. Poly(carboxybetaine methacrylate) grafted on PVA hydrogel via a novel surface modification method under near-infrared light for enhancement of antifouling properties[J]. Colloids and Surfaces A Physicochemical and Engineering Aspects, 2021, 617:126369.

The Line 231-232 was revised as following:

Fig. 7 showed the typical differential scanning calorimeter (DSC) curves of ul-trafine fibrous FEP porous membranes samples and the corresponding datas were tab-ulated in Table 1. As shown in the heating curves [Fig. 7(A)], there was a endothermic peak at 87.5℃ of nascent ultrafine fibrous FEP/PVA membranes. Moreover, the endo-thermic peak disappeared after sintering. These results indicated that the PVA was to-tally decomposed during the sintering process. The melting temperature of ultrafine fibrous FEP porous membranes increased with the increase of sintering temperature. Meanwhile, the enthalpy and the degree of crystallinity (Xc) increased. As for cooling curves [Fig. 7(B)], the crystallization peak moved towards the lower temperatures with the increase of sintering temperature. These results should be attributed to the nascent electro-spun fibers were randomly distributed and not interconnected. Sintering pro-cess enhanced the dimensional integrity and mechanical properties of the membranes.

Figure 7. DSC curves of ultrafine fibrous FEP porous membranes at different sintering temperature (FEP/PVA mass ratio: 1:6; a: nascent membrane; b:260℃; c: 280℃; d: 300℃; e:320℃; A-heating curve, B-cooling curve).

  1. Comment: Conclusion

Line 341 – When discussing “optimal conditions”, please refer to the context.

Response: It is really true as Reviewer suggested that when discussing “optimal conditions”, the context needed to be referred. The revised details was as following:

The optimal conditions were chosen that the FEP/PVA mass ratio was 6:1 and sintering temperature was 300℃ for 10 minutes, comparing with other preparation conditions.

Finally, special thanks to you for your good comments.

We have tried our best to improve the manuscript and made some changes in the manuscript. We appreciate for your warm work earnestly and hope that the correction will meet with approval. Thanks a lot for your comments and suggestions.
